# La-MAML: Look-ahead Meta Learning for Continual Learning, ML Reproducibility Challenge 2020

## Reproducibility Summary

The Continual Learning (CL) problem involves performing well on a sequence of tasks under limited compute. Current algorithms in the domain are either slow, offline or sensitive to hyper-parameters. La-MAML, an optimization-based meta-learning algorithm claims to be better than other replay-based, prior-based and meta-learning based approaches.

### Scope of Reproducibility

According to the MER paper [2], metrics to measure performance in the continual learning arena are Retained Accuracy (RA) and Backward Transfer-Interference (BTI). La-MAML [1] claims to be a better performing, robust and faster algorithm compared to the existing baselines. These are the main claim of the paper.

### Methodology

We used the author's code which was pretty new and built on the latest packages. Most of the experiments were tried on Free Kaggle Notebooks (Tesla P100 GPU). We ran the code according to the hyperparameters given in the original paper. We found that the results were very similar to the ones given in the paper.

### Results

We reproduced the Retained Accuracy on real world datasets to within 6% of the reported value, which supports the paper's conclusion that it outperforms the baselines.

### What was easy

Running the code was easy. The packages used for the official implementation were the latest. It was easy to incorporate Weights and Biases into the implementation.

### What was difficult

For some of the experiments, the computational requirement was too high. For example, the MNIST Many Permutations Dataset requires more than 12GB of RAM to pass into the loader. Further, some other experiments exceeded 12 hours of running time due to which we had to use less powerful GPUs.

### Communication with original authors

For most of the experiments concerning the main claim of the paper, the code was enough from the official repo provided by the authors on Github. However, reproducing some of the figures and the tables involving Gradient Alignment and Catastrophic Forgetting visualization proved to be difficult due to those parts not being published. We were able to contact the authors and received help for those experiments.

Preprint. Under review.

# 1  Introduction

In continual learning, the main task is to learn an unknown number of sequentially arriving tasks $T_1, \cdots, T_K$ in a supervised setting. In standard supervised learning, gradient descent does a good job of finding the minimum of one task. However, in continual learning, there is an issue of catastrophic forgetting, where a model forgets previous tasks that it has learned.

In particular, if a classifier trained on $T_1$ is then trained on the training data for task $T_2$, the catastrophic forgetting phenomenon demonstrates that its accuracy on $T_1$ will drop. In this way, the model unlearns what it has learned on task $T_1$.

Catastrophic forgetting results because the gradient of the model on the data for task $T_1$ may not necessarily be aligned with the data for task $T_2$. In the extreme case, where the gradients are perfectly unaligned, the model would be expected to forget everything it had previously learned.

Given infinite memory, this might be a trivial problem, as one could simply store all the training data for previous tasks. However, the challenge posed in continual learning is that the amount of memory given is limited. Various solutions have been developed to solve this problem, such as using a replay buffer to store previous samples, reserving a subset of neurons for each task and slowly growing the network as new tasks are seen, and storing previous gradient directions and making sure new updates are approximately aligned with them.

La-MAML is a meta-learning approach to continual learning that uses mechanisms from all three solutions. La-MAML was shown to perform better than Meta Experience Replay (MER) algorithm (previous state-of-the-art) on standard continual learning metrics, retained accuracy (RA) and backward-transfer and interference (BTI).

La-MAML consists of an inner level and an outer level of optimization. In the inner level, we start at some meta-parameters $\theta_0$ and try to minimize the task-specific loss function $l_t$ via $k$-steps of SGD, arriving at $\theta_k$. This can roughly be expressed as follows:

$$\theta^k \approx \arg\min_\theta l_t(\theta).$$

In the outer level, the goal is to find a good initialization $\theta_0$, doing so by measuring the loss of $\theta_k$ on the task-general loss function $L_t = \sum_{i=1}^t l_i$, roughly

$$\theta_0 \approx \arg\min_\theta L_t(U_k(\theta))$$

where $U_k$ denotes $k$ steps of gradient descent on the loss function $L_t$.

# 2  Terms and Definitions

In this section, we introduce a few terms and definitions that are used in this paper.

**Retained Accuracy (RA):** average accuracy of the model across all the tasks at the end of training

**Backward Transfer and Interference (BTI):** For each task, we calculate the difference in accuracy on that task right after that task was just learned and at the end of the entire training procedure. The BTI is the average of these values.

Now, we introduce some of the algorithms, both from previous papers and the current paper:

**Experience Replay (ER):** ER is an algorithm that stores a small replay buffer to store old data using reservoir sampling, ensuring that each sample has an equal probability of being in the buffer after it is seen. The buffer is then used to remember old tasks in future training.

**La-ER:** La-ER is an algorithm used in ablation study where parameter updates are identical to those used in experience replay (using the replay buffer), but the learning rate is varying throughout training and is updated in the same way that La-MAML is. By comparing La-ER to ER, we can see the benefits of tuning the learning rate according to the gradients during training.

**C-MAML:** C-MAML is the base algorithm used by La-MAML where the learning rate of both the inner and outer loops are fixed. The difference between C-MAML and La-MAML is that in La-MAML, the step size $\alpha$ is updated during each meta-iteration and is used for both inner and outer loops.

**Sync:** Sync is a hybrid between C-MAML and La-MAML where the learning rate for the outer loop is fixed, but the learning rate for the inner loop is updated as in La-MAML. THe difference between this and La-MAML is that in La-MAML, the outer loop and inner loop step sizes are both updated via gradient descent.

Further the experiments are conducted on two types of setups, distinguished based on how the incoming task datasets can be used by the model during training:

**Single Pass:** In this setup, once processed, data samples are not accessible anymore unless they were added to a replay memory.

**Multiple Pass:** In this setup, we can re-train the model using the already seen data over and over many times. This means that it is easier to get higher performance in this setup, compared to the Single Pass setup where data can only be used once.

We describe some terms associated with La-MAML:

**Asynchronous Updates:** It refers to the idea that the same inner learning rate (which gets modulated during each outer loop) is used for the outer loop meta-update also. This is in contrast to the synchronous updates of algorithms such as Meta-SGD, where the outer learning rate is fixed, evenwhen the inner learning rate is getting modulated.

**Sample Efficient Objective:** La-MAML's objective function (as described in the original paper) is different from that of MER. La-MAML's objective function tries to align the gradients of the current task with the average of old tasks, while MER's objective function tries to align the gradient of the current task pairwise with gradients of all the old tasks. La-MAML makes use of the new research that their technique is enough to obtain good results. This makes MER much slower in training time.

## 3    Scope of reproducibility

Main Claims of the paper:

1. La-MAML achieves better RA/BTI values than MER and other baselines, on real-world visual classification benchmarks.

2. La-MAML is faster than MER.

3. Learnable LRs and the asynchronous update of La-MAML are especially responsible for the increase in performance.

Additional Claims:

1. The performance increase obtained for La-MAML increases as we move from toy datasets such as MNIST to real world datasets such as CIFAR and Imagenet.

2. La-MAML is more robust to hyperparameter tuning than its variants.

## 4    Methodology

We used the author's code for most of the experiments. The hyperparameters were given in their paper. We only had to reoptimize hyperparameters for a few experiments.

All of our experiments have been run on free compute that we could get access to. For bug fixes and software developments involving less than an hour, we used Google Colab Notebooks (Tesla T4 GPU). For most of the experiments in our reproduced results, we used Kaggle Notebooks (Tesla P100 GPU). For a few experiments which take more than 9 hours (maximum limit of Kaggle Notebooks), we used the GPUs provided by our college (Tesla P4 GPU Cluster). And finally for experiments that require more than 20GBs of RAM such as the ones on the MNIST datasets, we used the 10 hour GPUs provided for free by Codeocean (Tesla K80 GPU).

Since most of the results were very close to that of the original paper, we only needed to reoptimize hyperparameters for a few experiments. We also verify the hyperparameter robustness patterns for the La-MAML variants in different settings than given in the original paper.

We used Weights and Biases to keep track of our experiments. It is very easy to incorporate and provides free cloud storage to make the results of our experiments publicly available.

### 4.1    Datasets

**MNIST Rotations**

Table 1: Datasets

|  | ROTATIONS | PERMUTATIONS | MANY | CIFAR-100 | TINYIMAGENET |
|---|---|---|---|---|---|
| No. of Tasks | 20 | 20 | 100 | 20 | 40 |
| No. of samples per task | 1000 | 1000 | 200 | 2500 | 2500 |
| No. of classes per task | 10 | 10 | 10 | 5 | 5 |
| Total no. of samples | 20,000 | 20,000 | 20,000 | 50,000 | 100,000 |

It is a variant of the MNIST dataset of handwritten digits [3], where each task contains digits rotated by a fixed angle between 0 and 180 degrees.

### MNIST Permutations

[5] It is a variant of MNIST, where each task is transformed by a fixed permutation of pixels. In this dataset, the input distribution for each task is unrelated.

### MNIST Many Permutations

It is a variant of MNIST Permutations that has 5 times more tasks (100 total) and 5 times less training examples per task (200 each).

### CIFAR-100

Incremental CIFAR100 [6], a variant of the CIFAR object recognition dataset with 100 classes [7], where each task introduces a new set of classes

### TinyImageNet-200

Tiny Imagenet is a scaled down version of ImageNet dataset [8].

## 4.2   Architecture

La-MAML uses the same architecture and experimental settings as in MER [2], allowing it to compare directly with their results. La-MAML uses the cross-entropy loss as the inner and outer objectives during meta-training.

Table 2: La-MAML Architecture for MNIST Datasets

Linear(784, 100)
ReLU()
Linear(100, 100)
ReLU()
Linear(100, 10)

## 4.3   Pseudocodes

| Theory | Implementation |
|---|---|

Theory:

**for** $t := 1$ **to** $T$ **do**
   **for** $ep := 1$ **to** $num_{epochs}$ **do**
      **for** batch $b$ **in** $(X^t, Y^t) \sim D_t$ **do**
         $k \leftarrow sizeof(b)$
         $b_m \leftarrow Sample(R) \cup b$
         **for** $n = 0$ **to** $k - 1$ **do**
            Push $b[k']$ to R with reservoir sampling
            $\theta_{k'+1}^j \leftarrow \theta_{k'}^j - \alpha^j \cdot \nabla_{\theta_{k'}^j}$
         **end for**
         $\alpha^{j+1} \leftarrow \alpha^j - \eta \nabla_{\alpha^j} L_t(\theta_k^j, b_m)$
         $\theta_0^{j+1} \leftarrow \theta_0^j - max(0, \alpha^{j+1}) \cdot \nabla_{\theta_0^j} L_t(\theta_k^j, b_m)$

Implementation:

```
for task in task_loader:
    for ep in num_epochs:
        for X,Y in task:
            for glance in glances:
                model_clone = model.clone()
                bx, by = (X,Y) + sample(Memory)
                for x,y in X,Y:
                    model_clone.inner_update(x, y)
                    meta_loss += lossfn(model_clone(bx), by)
                learning_rate.update(meta_loss)
                model.meta_update(meta_loss, learning_rate)
```

### 4.4 Hyperparameters

We used the default hyperparameters that were given in the paper. The authors however used a gridsearch to arrive at their values.

### 4.5 Experimental setup and code

We provide clear steps to setup the experiment on various free platforms such as Codeocean, Google Colab, or Kaggle, in our Github repo.

We include a description of how the experiments were set up that's clear enough a reader could replicate the setup. The measure used to evaluate the code is the Final Test Accuracy and the BTI values. Link to our code - Github

We also make publicly available all our experimental runs, which include the various metrics, logs, hyperparameters and even variations in loss values, in our Weights and Biases project page - WandB

### 4.6 Computational requirements

We have indicated the time required to run each experiment in the Results tables. Due to the limitations of Free-GPUs, we ran our experiments mainly on three different type of systems:

1. Kaggle - Tesla P100 GPU. It gives 40 hours of GPU per week, with maximum 9 hours of continuous usage.

2. Institute GPU - Slowest. Tesla P4 GPU. It gives unlimited continuous usage. Provided by the Indian Institute of Technology for its students only.

3. Codeocean - Tesla K80 GPU. It gives total 10 hours of GPU usage per month. More than 12 GB of RAM is available.

Most of the experiments were done on Kaggle Notebooks. Some experiments required more than 9 hours of continuous running time, for which we used the Institute GPU. The MNIST Many Permutations experiment requires more than 12GBs of RAM, due to which we used Codeocean for running the experiments.

The total compute time we consumed after all the experiments for this report, were 48 days (*24 hours). It has to be taken into consideration that we ran a lot of the experiments in parallel, so it did not take 48 days for us while running the experiments. If you only have access to free GPUs, running all these will be very difficult because the lack of continous running time means that you have to spend a lot of time consistently over days being vigilant on running the next set of experiments soon after the previous ones are completed.

## 5 Results

Our results confirm the main claims of the paper. We were able to reproduce metrics close to 6% of the original paper.

### 5.1 Results reproducing original paper

#### 5.1.1 Result 1: Performance on MNIST Datasets

Table 3: Original Results

| METHOD | ROTATIONS | | PERMUTATIONS | | MANY | |
|---|---|---|---|---|---|---|
| | RA | BTI | RA | BTI | RA | BTI |
| ONLINE | $53.38 \pm 1.53$ | $-5.44 \pm 1.70$ | $55.42 \pm 0.65$ | $-13.76 \pm 1.19$ | $32.62 \pm 0.43$ | $-19.06 \pm 0.86$ |
| EWC | $57.96 \pm 1.33$ | $-20.42 \pm 1.60$ | $62.32 \pm 1.34$ | $-13.32 \pm 2.24$ | $33.46 \pm 0.46$ | $-17.84 \pm 1.15$ |
| GEM | $67.38 \pm 1.75$ | $-18.02 \pm 1.99$ | $55.42 \pm 1.10$ | $-24.42 \pm 1.10$ | $32.14 \pm 0.50$ | $-23.52 \pm 0.87$ |
| MER | $\mathbf{77.42} \pm \mathbf{0.78}$ | $\mathbf{-5.60 \pm 0.70}$ | $73.46 \pm 0.45$ | $-9.96 \pm 0.45$ | $47.40 \pm 0.35$ | $-17.78 \pm 0.39$ |
| C-MAML | $77.33 \pm 0.29$ | $-7.88 \pm 0.05$ | $\mathbf{74.54} \pm \mathbf{0.54}$ | $-10.36 \pm 0.14$ | $47.29 \pm 1.21$ | $-20.86 \pm 0.95$ |
| SYNC | $74.07 \pm 0.58$ | $-6.66 \pm 0.44$ | $70.54 \pm 1.54$ | $-14.02 \pm 2.14$ | $44.48 \pm 0.76$ | $-24.18 \pm 0.65$ |
| LA-MAML | $\mathbf{77.42} \pm \mathbf{0.65}$ | $-8.64 \pm 0.403$ | $74.34 \pm 0.67$ | $\mathbf{-7.60} \pm \mathbf{0.51}$ | $\mathbf{48.46} \pm \mathbf{0.45}$ | $\mathbf{-12.96} \pm \mathbf{0.073}$ |

Table 4: Reproduced Results

| METHOD | ROTATIONS | | PERMUTATIONS | | MANY | |
|---|---|---|---|---|---|---|
| | RA | BTI | RA | BTI | RA | BTI |
| C-MAML | 77.2±0.7 (5 mins) | -8.2±0.8 | 74.3±0.5 (5 mins) | -10.7±0.4 | 46.6±0.9 (10 mins) | -21.3±1.0 |
| SYNC | 71.3±0.6 (7 mins) | -14.9±0.5 | 59.0±0.7 (7 mins) | -19.5±0.8 | 43.7±1.3 (16 mins) | -23.8±0.7 |
| LA-MAML | 76.7±0.7 (6 mins) | -9.2±0.7 | 74.1±1.0 (5 mins) | -7.8±0.9 | 47.9±0.4 (16 mins) | -13.4±0.1 |

As mentioned earlier, higher RA and lower BTI is desirable. Except for the Sync-Rotations and Sync-Permutations result, the RA values we obtained have less than 2% deviation from that of the original paper, while the BTI values have less than 8% deviation. We used a minimum of 4 seeds for the experiments.
We do not reproduce ONLINE, EWC, GEM and MER experiments here, neither have the original authors, as the accuracy values given are extracted directly from the MER paper.

We note that La-MAML obtains a slightly lower RA (76.7) than from the original paper (77.42) in the Rotations dataset, using 6 seeds, which means it slightly underperforms compared to the previous baseline MER (77.42). Since the difference is less than 1% we assume this is due to lucky seeds used by the original authors. We should also note that MER requires 30 minutes of runtime to arrive at its RA compared to 5 minutes for La-MAML. We will also see from the next section (Results on Real World datasets) that La-MAML outperforms MER with a higher margin when moving to harder datasets.

Among the La-MAML variants, La-MAML and C-MAML have more or less similar performance, while Sync underperforms both all the time. Sync's underperformance proves the importance of the asynchronous update.
The MNIST Many Permutations dataset requires minimum 25GB of Disk Space and also uses over 20GBs of RAM while loading the dataset. This makes it unsuitable on most Free GPUs, except that of Codeocean.

**Hyperparameter Reoptimization:**
- For Sync Permutations, the default hyperparameters given in the original paper gave us a very low result (59.0). After hyperparameter reoptimization, we were able to get to a higher result of 67.9±1.7. This is still 4% lower than that given in the paper, but we noticed that the deviation in RA between seeds were large. The optimized hyperparameter values are 0.001, 0.06 and 0.01 for $\alpha_0$, $\beta$ and $\eta$ respectively. The meaning of the hyperparameters are described in section 5.1.5.
- Sync Rotations gave RA and BTI values that are 4% and 124% different, respectively. Due to the other experiments taking a large portion of our time and energy, we were unable to reoptimize the hyperparameters on time.

### 5.1.2 Result 2: Performance on Real World Datasets

Table 5: Original Results

| METHOD | CIFAR-100 | | | | TINYIMAGENET | | | |
|---|---|---|---|---|---|---|---|---|
| | MULTIPLE | | SINGLE | | MULTIPLE | | SINGLE | |
| | RA | BTI | RA | BTI | RA | BTI | RA | BTI |
| IID | $85.60 \pm 0.40$ | - | - | - | $77.1 \pm 1.06$ | - | - | - |
| ER | $59.70 \pm 0.75$ | $-16.50 \pm 1.05$ | $47.88 \pm 0.73$ | $-12.46 \pm 0.83$ | $48.23 \pm 1.51$ | $-19.86 \pm 0.70$ | $39.38 \pm 0.38$ | $-14.33 \pm 0.89$ |
| ICARL | $60.47 \pm 1.09$ | $-15.10 \pm 1.04$ | $53.55 \pm 1.69$ | **$-8.03 \pm 1.16$** | $54.77 \pm 0.32$ | **$-3.93 \pm 0.55$** | $45.79 \pm 1.49$ | **$-2.73 \pm 0.45$** |
| GEM | $62.80 \pm 0.55$ | $-17.00 \pm 0.26$ | $48.27 \pm 1.10$ | $-13.7 \pm 0.70$ | $50.57 \pm 0.61$ | $-20.50 \pm 0.10$ | $40.56 \pm 0.79$ | $-13.53 \pm 0.65$ |
| AGEM | $58.37 \pm 0.13$ | $-17.03 \pm 0.72$ | $46.93 \pm 0.31$ | $-13.4 \pm 1.44$ | $46.38 \pm 1.34$ | $-19.96 \pm 0.61$ | $38.96 \pm 0.47$ | $-13.66 \pm 1.73$ |
| MER | - | - | $51.38 \pm 1.05$ | $-12.83 \pm 1.44$ | - | - | $44.87 \pm 1.43$ | $-12.53 \pm 0.58$ |
| META-BGD | $65.09 \pm 0.77$ | $-14.83 \pm 0.40$ | $57.44 \pm 0.95$ | $-10.6 \pm 0.45$ | * | * | $50.64 \pm 1.98$ | $-6.60 \pm 1.73$ |
| C-MAML | $65.44 \pm 0.99$ | $-13.96 \pm 0.86$ | $55.57 \pm 0.94$ | $-9.49 \pm 0.45$ | $61.93 \pm 1.55$ | $-11.53 \pm 1.11$ | $48.77 \pm 1.26$ | $-7.6 \pm 0.52$ |
| LA-ER | $67.17 \pm 1.14$ | $-12.63 \pm 0.60$ | $56.12 \pm 0.61$ | $-7.63 \pm 0.90$ | $54.76 \pm 1.94$ | $-15.43 \pm 1.36$ | $44.75 \pm 1.96$ | $-10.93 \pm 1.32$ |
| SYNC | $67.06 \pm 0.62$ | $-13.66 \pm 0.50$ | $58.99 \pm 1.40$ | $-8.76 \pm 0.95$ | $65.40 \pm 1.40$ | $-11.93 \pm 0.55$ | **$52.84 \pm 2.55$** | $-7.3 \pm 1.93$ |
| LA-MAML | **$70.08 \pm 0.66$** | **$-9.36 \pm 0.47$** | **$61.18 \pm 1.44$** | $-9.00 \pm 0.2$ | **$66.99 \pm 1.65$** | **$-9.13 \pm 0.90$** | $52.59 \pm 1.35$ | **$-3.7 \pm 1.22$** |

Table 6: Reproduced Results

| METHOD | CIFAR-100 | | | | TINYIMAGENET | | | |
|---|---|---|---|---|---|---|---|---|
| | MULTIPLE | | SINGLE | | MULTIPLE | | SINGLE | |
| | RA | BTI | RA | BTI | RA | BTI | RA | BTI |
| IID | 84.8±0.3 (3 hrs) | | | | 77.2±0.8 (4 hrs) | | | |
| ER | 59.7±1.0 (30 mins) | -16.6±0.4 | 45.5±0.1 (10 mins) | -15.3±1.1 | 48.7±1.2 (3 hrs) | -18.9±0.9 | 38.1±1.0 (1 hrs) | -15.9±1.7 |
| ICARL | 60.9±0.9 (1 hrs) | -15.7±0.1 | 52.4±0.7 (10 mins) | **-8.7±0.5** | 54.8±0.4 (5 hrs) | **-4.0±0.2** | 48.4±0.5 (1 hrs) | **-3.6±0.6** |
| GEM | 61.1±0.6 (4 hrs) | -18.4±0.4 | 47.2±0.9 (30 mins) | -15.5±1.1 | 48.9±0.8 (6 ds)** | -21.6±0.1 | 42.6±0.6 (5 hrs) | -13.0±0.4 |
| AGEM | 54.9±2.1 (2 hrs) | -19.8±1.6 | 44.4±0.8 (30 mins) | -16.1±1.6 | 45.0±1.2 (5 ds)** | -21.3±0.5 | 40.3±0.4 (3 hrs) | -13.0±1.6 |
| MER | | | 51.4±1.2 (4 ds)** | -13.0±1.4 | | | 43.1±1.3 (9 ds)** | -13.1±0.7 |
| META-BGD | 65.7±1.6 (3 hrs) | -13.9±1.1 | – | – | – | – | – | – |
| C-MAML | 65.8±1.1 (3 hrs) | -13.1±1.2 | 55.9±1.1 (30 mins) | -9.2±0.3 | 60.7±1.3 (3 ds)** | -11.8±1.0 | 47.8±0.8 (2 hrs) | -7.6±0.8 |
| LA-ER | 67.6±1.0 (2 hrs) | -14.2±5.0 | 54.7±1.6 (10 mins) | -8.6±0.8 | 54.7±1.8 (5 hrs) | -14.0±1.2 | 43.8±1.6 (1 hrs) | -13.7±1.3 |
| SYNC | 67.6±1.8 (2 hrs) | -13.6±1.3 | 56.5±1.6 (1 hrs) | -10.6±1.2 | 64.0±1.5 (4 ds)** | -12.8±0.7 | **53.7±0.9 (2 hrs)** | -7.2±0.8 |
| LA-MAML | **71.0±1.0 (3 hrs)** | **-7.5±0.5** | **60.5±1.7 (2 hrs)** | -9.7±2.1 | **66.0±1.7 (3 ds)**** | **-10.3±0.8** | 51.7±1.2 (2 hrs) | **-2.7±0.7** |

** - indicates Institute GPU

Overall, The RA values are less than 6% different from the original paper. Only 9 results were above 4% deviation. We observe that there is 5 to 7% deviation in RA between the seeds of the same experiment, in many cases. The original results were only evaluated on 3 seeds. We did not feel it wise to spend our limited time and compute on hyperparameter tuning, since the deviation from the original paper could easily be due to lucky/unlucky seeds. Also, the deviations do not affect the main conclusions of the experiment (see the results marked in bold font).

The BTI values are roughly within 20% of the original paper. This is not very unexpected since the variances between different seeds of the same experiment are around 20%, in many cases.

In CIFAR-Single and TinyImageNet-Single setups, La-MAML has 17.7% and 20% improvement over MER, respectively.

We see that La-MAML and its variants consistently outperform the previous baselines, interms of RA values. We also see that, apart from MER, ICARL is a significant competitor. Although ICARL has better BTI in 3 of the 4 cases, La-MAML outperforms it in RA values each time.

**The Ablations:**

- La-ER (Lookahead-Experience Replay) outperformed its ancestor ER (Experience Replay) by 15% and 25% interms of RA and BTI values, on average. ER primarily relies on its replay buffer to remember old tasks. It doesn't have a meta-learning step implying it doesn't encourage gradient alignment between tasks that way. Hence, La-ER's superior performance testifies to the improvement obtained solely by having modulated learning rates. This also opens many lines of research where the benefits of modulated learning rates to other algorithms in the Continual Learning and Meta-Learning can be explored.

- The Meta-BGD experiments were very unstable. In CIFAR-Single and TinyImagenet-Single setups, it crashed to an RA of 20 (lowest possible RA in 5 way classification), in 2/3 of the seeds, hence we did not consider it meaningful to include it in our study.

Both the MER experiments exceeded 9 hours running time. Apart from MER, the running times exceeded 9 hours only for some experiments in the TinyImagenet-Multiple setup. Excluding these 7 long experiments, running all the others in this section will take (63 hours x 3 seeds) = 189 hours of compute on Kaggle GPUs (Tesla P100).

### 5.1.3   Result 3: Hyperparameter Robustness

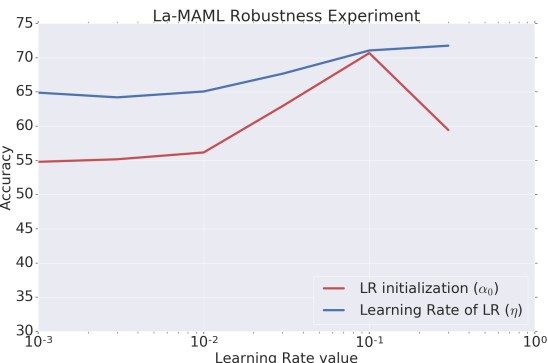

Figure 1: Original Result: CIFAR-Multiple Setup

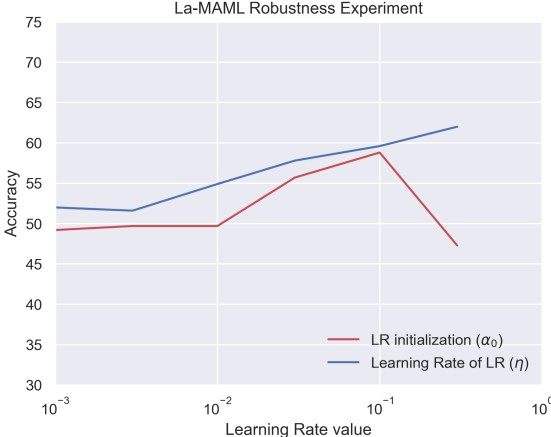

Figure 2: Reproduced Result: CIFAR-Single Setup

Figure 2 and 3 are the Retained Accuracy (RA) vs Learning Rate (LR) plots for the La-MAML algorithm. They are plotted by varying one hyperparameter while keeping others fixed at their optimal value. The hyperparameters are varied between [0.001, 0.3]. The optimal values for LR Initialization ($\alpha_0$) and Learning Rate of LR ($\eta$) are 0.25 and 0.1 respectively.

Although both plots are obtained for the CIFAR-100 dataset, Figure 2 is obtained for the Multiple Pass setup while Figure 3 is obtained for the Single Pass setup. The differences in the setups are described in the section 2. As mentioned, Figure 2 is the original result from the paper while Figure 3 is what we reproduced. Our plot is made from the average of three seeds. We use the test accuracy values, not the train accuracy values for the plot.

We can notice the similarity in shape between the two plots. However, we can see that the RA values for Figure 3 (ours) is lower than that for Figure 2. This is expected because, as mentioned in the section 2, Single Pass is more difficult and hence the lower accuracy values. We did not use the Multiple Pass setup (like in the original paper) because of compute and resource constraints. The Single Pass setup takes roughly half the running time to that of Multiple Pass.

**Claim from the paper:** La-MAML is robust to hyperparameters.

**How its verified here:**

Note that the RA while tuning Learning Rate of LR ($\alpha_0$) is steadily increasing, but that increase is gradual since across a wide range of LRs varying over 2 orders of magnitude (from 0.003 to 0.3) the RA increases from 51.6 to 62, the difference in RA being 17% ((62 - 51.6) / 62).

In the original paper, for the multiple pass setup, this difference in RA (while tuning $\eta$) between highest and smallest RA is claimed to be only 6%. However if we look closely at its graph, we can see that the the maximum and minimum values of RA are roughly 72% and 64%, which makes the difference to be about 11% ((72 - 64) / 72). This might have been due to the original authors using a different method to calculate the percentage or may even be an error, but raising this confusion helps put our 17% in correct perspective. The difference between the 17% deviation of ours and 11% of theirs can be explained by the fact that the single pass setup is more difficult implying that the hyperparameters have more impact on the performance. This idea is further reinforced in the next section where hyperparameter tuning is done on the simpler MNIST dataset. We also note that 11% or 17% may not be considered trivial, and hence we cannot corroborate with the original paper's claim that the RA deviation while tuning $\eta$ can be ignored.

The RA values we get while tuning $\eta$ are [52.0, 51.6, 55.0, 57.8, 59.60, 62.0]. From Table 6 we see that highest among the previous baselines (MER) has RA of 51.4. So, even without tuning $\eta$, La-MAML outperforms previous baselines.

#### 5.1.4 Result 4: Running Times

Table 7: Running times (in minutes) for MER and La-MAML on MNIST benchmarks

| METHOD | ROTATIONS |
|---|---|
| MER | 30.15 ± 1.17 |
| LA-MAML | 5.84 ± 0.18 |

From the above table, we see that MER is roughly 6 times slower than La-MAML on the MNIST Rotations Dataset.

We can also observe from Table 6 that, for the Single Pass setups in CIFAR-100 and TinyImageNet Datasets, MER (53.2 and 43.1) exceeded 9 hrs of runtime while La-MAML (60.5 and 51.7) achieved a higher accuracy in just 2 hours. This indicates the fastness of La-MAML.

#### 5.1.5 Result 5: Hyperparameter Robustness of La-MAML, Sync and C-MAML

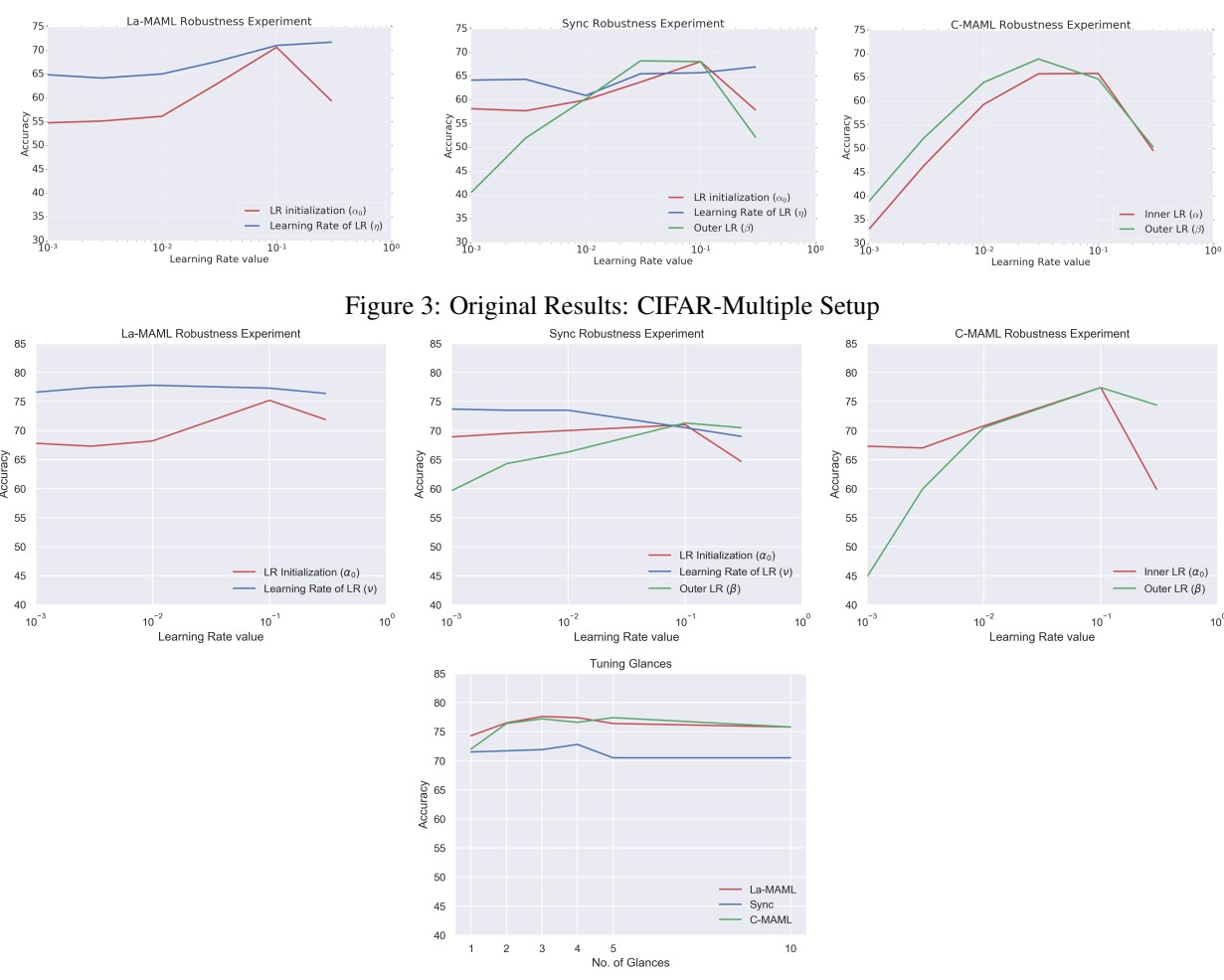

Figure 3: Original Results: CIFAR-Multiple Setup

Figure 4: Reproduced Results: MNIST-Rotations Setup

In this section we analyse the robustness of the La-MAML variants to their LR-related hyper-parameters on the MNIST Rotations dataset. Our three variants have different sets of these hyper-parameters which are specified as follows:

- C-MAML: Inner and outer update LR (scalar) for the weights ($\alpha_0$ and $\beta$)

- Sync La-MAML: Inner loop initialization value for the vector LRs ($\alpha_0$), scalar learning rate of LRs ($\eta$) and scalar learning rate for the weights in the outer update ($\beta$)

- La-MAML: Scalar initialization value for the vector LRs ($\alpha_0$) and a scalar learning rate of LRs ($\eta$)

The first figure contains the results of hyperparameter tuning on the CIFAR-100 dataset in the Multiple Pass setting, while the second figure represents the results on the MNIST Rotations dataset.

In the original result, we can notice that the RAs vary more widely while modulating the hyperparameters, than in the reproduced result. This is likely because the MNIST dataset is simpler than the CIFAR-100 dataset. Performance varies less widely in simpler datasets because it is easier to reach the minima of a less complicated loss surface.

We can also note that the difference in performance with tuning Learning Rate of LR ($\eta$) for La-MAML is very small even with 2 orders of magnitude change in $\eta$ from 0.0001 to 0.3. In Sync, only the change in Outer LR ($\beta$) seems to matter, other hyperparameters give roughly same RA. In C-MAML, both the hyperparameters matter very much.

Glances hyperparameter indicates the number of gradient updates or meta-updates made on each incoming sample of data. In the Single-Pass setup, it becomes essential to take multiple gradient steps on each sample (or see each sample for multiple glances), since once we move on to later samples, we can't revisit old data samples. As for glances, we notice that there isn't much variance in the RA values even with large changes in the number of glances. This may be because we were using a simpler dataset, hence drowning out any significant patter. We observe a spike in RA at glances 5, for La-MAML and C-MAML, while a spike at 4 for Sync. 5 glances are the default value given in the original paper.

All RA values are calculated using the mean of 2 seeds.

### 5.2 Additional Result: Reimplementing part of the pipeline

We were able to replace the inner loop and outer loop updates by the Meta-SGD (a Meta-Learning algorithm which uses per-parameter learning rates which are synchronously updated) algorithm. We obtained an RA of 70 (average of 3 seeds) in the MNIST Rotations dataset, with very little hyperparameter tuning. If you look at section 5.1.1, you can see that an RA of 70 is less than that of the La-MAML variants and MER, but greater than the other baselines. Sadly, we were not able to do many experiments with this, due to the deadline of the paper submission, but it does give us possibilities of easily substituting the inner mechanisms of La-MAML with mechanisms from the Meta-Learning literature to adapt to different problem arenas in Continual Learning. We publish our code on Github - Link.

## 6 Discussion

From our results, we make the following observations:

- La-MAML is a scalable, fast and robust algorithm outperforming previous baselines in the continual learning arena. It is scalable meaning that it achieves higher margins of improvement on real world datasets.
- La-MAML has two mechanisms: (1) modulated per-parameter learning rates & (2) asynchronous meta-updates, which are individually responsible for its high performance. This is verified by the ablation studies on La-ER, Sync and C-MAML in sections 5.1.1 and 5.1.2. Thirdly, La-MAML's sample efficient objective is responsible for its shorter running time.
- Among the La-MAML variants, La-MAML achieves high accuracy while requiring less hyperparameter tuning.

In general, we notice a trend of having high variance in results, between the different seeds of the same experiment, in all the various continual learning algorithms. The ablation experiments show that the individual mechanisms introduced in the paper can be used individually in combination with other algorithms in related fields of meta-learning and multi-task learning.

### 6.1 What was easy

Since the author's published the code to create Tables 1 and 3, it was fairly straightforward to incorporate wandb and collect all the experimental data.

Additionally, the hyper parameters were given in the paper clearly which meant that we didn't have to spend time on tuning most of them, except a few anomalies which we have described in the respective sections of the report. The results matched closely to that of the paper easily.

## 6.2 What was difficult

The data-loader for MNIST Many Permutations require more than 12GBs of RAM. This is not available in most free GPUs or even in our laptops. Only Codeocean provides this, but its only for 10 hours, which limits the extent of experimentation.

Although most of the experiments, with a few exceptions, can be performed individually on Free GPUs, although it will be a very time consuming (multiple weeks) process to run all the experiments in the paper, in series. The few exceptions which exceed 9 hours of running time require access to paid GPUs or University GPU clusters, like in our case, where experiments can be run continously over long periods.

The code to calculate the Gradient Alignment had not been published. Although a description of how to calculate it had been given in the MER paper, it left some details ambiguous, due to which we were unable to reproduce it on time.

## 6.3 Communication with original authors

We were able to contact the authors and received help for some additional experiments not given in the official code repository.

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
