# OpenReview forum: "La-MAML: Look-ahead Meta Learning for Continual Learning, ML Reproducibility Challenge 2020"
_ML_Reproducibility_Challenge/2020 — Reject_

### Official Review · AnonReviewer3 · 2021-02-09
**A successful reproduction experiment but with limited insights**

**Rating:** 3
**Confidence:** 4

**Review:**

This paper tries to reproduce the ICLR2019 paper "Learning to learn without forgetting by maximizing transfer and minimizing interference," by Matthew Riemer et al.

The authors present an straightforward 1to1 reproduction of the original paper experimental setting. They ran the code according to the instructions given in the official implementation and got very similar results. This is OK for the the ML Reproducibility Challenge. However, the authors failed to provide meaningful results/insights for a majority of the analysis performed. In this regard, many components of the proposed results, ablation and running times studies in the manuscript are incomplete or limited, without offering clear descriptions and/or insight into the outcomes.


Itemised comments:

- Introduction is too short (seems a bullet list) and non-informative about the problem addressed, its importance, motivation, etc. Also, La-MAML should be better introduced and described.
- Repetitiveness: Methodology and Computational requirements sections are mostly the same. Also Section 5.2.
- Pseudocode in Section 3.3 shouldn't be a low-resolution figure.
- Tables 3 and 5 shouldn't be low-resolution figures. Lack of consistency wrt other tables.
- Sections 4.1.3 and 4.1.4 placed after section 5.
- Ablation analyses are not explained. The reader cannot get any useful insight from section 5.
- Results and additional results are vaguely described. This makes it very hard for the reader to follow their interpretation.



**Familiar With The Original Paper:**

I have not read the original paper

**Reproducibility Summary:**

Report has summary

---

### Official Review · AnonReviewer2 · 2021-03-02
**Shows the original experiments are repeatable with original code, but does not add much**

**Rating:** 4
**Confidence:** 4

**Review:**

This report repeats a large number of experiments from the original article, using the original code, confirming the claims in scope. The authors spent time and ingenuity adapting and packaging the code to run on 4 different environments, to satisfy runtime and memory requirements of different experiments. However, repeating the original experiments does not shed much light on the original paper, and this work does feature re-implementation, new experiments, ablation studies, or hyper-parameter re-optimization. The small discrepancies that appear in some figures are not discussed.
As the [task description](https://paperswithcode.com/rc2020/task) of the challenge says, "Just re-running code is not a reproducibility study", and this submission does not provide additional insight on the original code or paper.

Reproducibility summary
--------------------------------------
The summary is filled up correctly, and summarizes well the report, although there are inconsistencies:
- the "Scope of Reproducibility" paragraph mentions only 1 claim, when Section 2 mentions 3, though they broadly overlap
- the "Methodology" paragraph only mentions one runtime environment (Google Colab with T4 GPUs), when Section 3 (Methodology), especially 3.6 (Computational requirements), expresses the need for using 4 different systems and says most experiments were using a different one (Kaggle Notebook with P100 GPU).

Scope of reproducibility
---------------------------------
The claims are clearly stated, and reproduced results connected to them. There is not much justification of how the results support the claims, except for the explanations in the original article.

Code and experiments
----------------------------------
The authors re-used code and hyper-parameters from the original repository.
No additional hyper-parameter search was reported.
No additional ablation studies were reported.
The report provides timing information for all experiments, and discusses speed differences and trade-offs of different systems. It could have been more clear which reported durations were obtained in which setting (Table 6 tags some results with "\*\*", and Section 3.6 suggests it may be "3. Institute GPU", but it's not mentioned in the caption).

Discussion on results
------------------------------
The submission reports accuracies "within 4%" of the original values, but does not elaborate on which experiments were further off from the originals, or how that "4%" (presumably 4 percentage points) was computed.
In fact:
- is not so meaningful when reported values for BTI can be "-2.73 ± 0.45", and
- In Table 3/4, the RA metric for "Sync" on "Permutations" went from "70.54 ± 1.54" to "59.3 ± 2.3"

Most of the results do seem to line up, and consistent with the provided confidence intervals, and there are reasonable explanations (one bad run can become likely when running many experiments, the original authors might have had a lucky random seed, this might go away when re-tuning hyper-parameters...), but this was not acknowledged or discussed in any way.

Similarly, Figure 3 shows a similar effect of the learning rate value as Figure 2 (original), but the whole graph is 5 to 10 % lower in absolute accuracy, without it being acknowledged or explained.




**Familiar With The Original Paper:**

I have not read the original paper

**Reproducibility Summary:**

Report has summary

---

### Decision · Program_Chairs · 2021-03-31

**Decision:**

Reject

**Comment:**

Overall reviews and/or the paper content not good enough for the AC to recommend to the journal.